

# Recent and rapid reef recovery around Koh Phangan Island, Gulf of Thailand, driven by plate-like hard corals

Florian Stahl[1,2,*], Selma D. Mezger[2,*], Valentina Migani[3], Marko Rohlfs[4], Victoria J. Fahey[5], Eike Schoenig[5] and Christian Wild[2]

[1] Faculty of Biology and Chemistry, Marine Botany Group, Universität Bremen, Bremen, Germany
[2] Faculty of Biology and Chemistry, Marine Ecology Group, Universität Bremen, Bremen, Germany
[3] Faculty of Biology and Chemistry, Evolutionary Biology Group, Universität Bremen, Bremen, Germany
[4] Faculty of Biology and Chemistry, Chemical Ecology Group, Universität Bremen, Bremen, Germany
[5] Center for Oceanic Research and Education (COREsea), Chaloklum, Koh Phangan, Thailand
* These authors contributed equally to this work.

Corresponding author
Selma D. Mezger,
mezger@uni-bremen.de

## ABSTRACT

Mass bleaching events and local anthropogenic influences have changed the benthic communities of many coral reefs with pronounced spatial differences that are linked to resilience patterns. The Gulf of Thailand is an under-investigated region with only few existing datasets containing long-term developments of coral reef communities using the same method at fixed sites. We thus analyzed benthic community data from seven reefs surrounding the island of Koh Phangan collected between 2014 and 2022. Findings revealed that the average live hard coral cover around Koh Phangan increased from 37% to 55% over the observation period, while turf algae cover decreased from 52% to 29%, indicating some recovery of local reefs. This corresponds to a mean increased rate of coral cover by 2.2% per year. The increase in live hard coral cover was mainly driven by plate-like corals, which quadrupled in proportion over the last decade from 7% to 28% while branching corals decreased in proportion from 9% to 2%. Furthermore, the hard coral genus richness increased, indicating an increased hard coral diversity. While in other reefs, increasing live hard coral cover is often attributed to fast-growing, branching coral species, considered more susceptible to bleaching and other disturbances, the reefs around Koh Phangan recovered mainly *via* growth of plate-like corals, particularly of the genus *Montipora*. Although plate-like morphologies are not necessarily more bleaching tolerant, they are important for supporting reef fish abundance and structural complexity on reefs, aiding reef recovery and sturdiness. Hence, our findings indicate that the intensity of local stressors around Kho Phangan allows reef recovery driven by some hard coral species.

## INTRODUCTION

Coral reefs belong to the most threatened ecosystems and are degraded worldwide (*Carpenter et al., 2008*). Global climate change (*Bellwood et al., 2004*; *Hoegh-Guldberg et al., 2007*; *Hughes et al., 2018b*; *Hughes & Connell, 1999*) and more local stressors, such as overfishing (*Jackson et al., 2001*), pollution (*Dubinsky & Stambler, 1996*) and physical damage (*Cheal et al., 2017*) are some of the threats coral reefs are facing. Coral reefs around the world are currently experiencing bleaching events on average every 6 years, a significant increase from the 1980s when bleaching occurred about once every 20–30 years (*Hughes et al., 2018a*). The increasingly short intervals between bleaching events combined with other stressors can impede the recovery of many reefs (*Carilli et al., 2009*; *Hughes et al., 2007*). On the habitat scale, it is important to determine the growth morphology of coral colonies in the assemblage, as they contribute to different ecosystem properties, such as habitat complexity (*Richardson, Graham & Hoey, 2017*), which further determines features like microhabitat availability (*Graham & Nash, 2013*) or larval recruitment (*Hata et al., 2017*). Coral growth form and size generally affect demographic rates such as fecundity (*Álvarez-Noriega et al., 2016*), growth (*Dornelas et al., 2017*), bleaching susceptibility (*McCowan, Pratchett & Baird, 2012*) or background mortality (*Madin et al., 2014*). The decline in hard coral cover in reefs can lead to so-called phase shifts. Reefs, which were usually dominated by hard corals have turned to a macroalgal dominance (*Hughes, 1994*; *Nyström et al., 2008*). In most cases, these shifts cannot be reversed (*Graham et al., 2013*). Reefs that have gone through phase shifts have reduced biodiversity and cannot provide as many ecosystem services (*Edinger et al., 1998*). The mentioned stressors and subsequent consequences are highly visible in the Gulf of Thailand. As most urban areas have poorly developed sewage treatment facilities or fisheries management measures, eutrophication (*Cheevaporn & Menasveta, 2003*) and overfishing (*Pauly & Chuenpagdee, 2003*) are severe problems in the whole Gulf of Thailand. A study comparing monitoring data collected with manta-tow surveys in 1995, 2006, and 2010 in the Gulf of Thailand showed that off-shore reefs were less impacted and threatened than those in near-shore waters, indicating a high anthropogenic influence from the coast (*Phongsuwan et al., 2013*).

Due to the slow growth of hard corals and the long time it takes to build and form a reef, it is important to understand long-term developments in hard coral benthic cover. There are only a few long-term studies for regions such as the Caribbean (*Contreras-Silva et al., 2020*; *Gardner et al., 2003*) and the Australian Great Barrier Reef (*De'ath et al., 2012*). Most of these studies were either meta-analyses or focused on large areas and low temporal resolution. Few studies were conducted with a focus on smaller areas or on individual reefs. Yet, in times of fast-changing climatic conditions and detrimental anthropogenic influences, it is necessary to have access to long-term data to be able to detect changes and trends. This is especially important for prompt management decisions and adjustments. One challenge we often face when using long-term data, is the comparability of the data. This can be a problem because data that is collected from different sources and monitoring programs may be gathered using different methodologies. For example, the data may be

collected at different temporal and spatial resolutions, which can make it difficult to accurately compare the data. Having data collected by one single organization ensures a concise method and reduces observer bias by having the same people over time controlling the collection and recording of data. Therefore, this study presents valuable data for a better understanding of long-term changes in coral cover in the Gulf of Thailand as a consistent survey method at fixed sites that have been utilized throughout a decade by one single organization.

The Center for Oceanic Research and Education (COREsea), situated on the island of Koh Phangan in the Gulf of Thailand, started its monitoring of local reefs in 2012. Data were collected on the coral coverage and community composition of reefs surrounding the island. We utilized a unique data set of 692 transect surveys from seven reefs around the northwest coast of Koh Phangan to answer the questions (1) how have the benthic reef communities, particularly hard coral cover, around Koh Phangan developed since 2014; (2) how have hard coral communities, in terms of growth forms and diversity, changed over time; and (3) have reefs around Koh Phangan undergone shifts from coral to algae dominance or *vice versa*.

## METHODS

### Study site

This study was carried out on coral reefs adjacent to the island of Koh Phangan (9°43′N 100°0′E), located in the lower Gulf of Thailand, approximately 60 km north-east of the mainland (Fig. 1). With a size of 125 km², Koh Phangan is the second largest island of the Samui-Archipelago.

The seven surveyed reefs were located along the northwest coast of the island. The sites were chosen due to their accessibility, similarity in reef structure, and representation of the main bays along the northwest of the island, encountering different impacts of tourism. The sites were all connected to beaches which were directly accessible through hotels, resorts, or public roads. All sites were frequently exposed to boat traffic and diving or snorkeling tourists. Mae Haad and Ao Chaloklum represented the largest bays, while Ao Thong Lang and Haad Khom were approximately half their area. Boat traffic was highest at Ao Chaloklum as its bay area and pier served as the main fishing port of the North coast of the island. Mae Haad was the reef which had the highest number of tourists, whereas Ao Thong Lang was smaller and due to its poorer road accessibility, less frequently visited. Further information about the different sites can be found in the Supplementary Material.

### Data collection

To document the development of live coral cover and evaluate benthic community composition, underwater surveys were conducted by SCUBA divers during COREsea's regular monitoring program. Surveys were conducted from 2012 to February 2020, with a pause due to Covid-19, and resumed in 2022. All transects were positioned on the reef crest between 4 and 6 m water depth depending on the tidal patterns and time of year. Detailed information about the amount of transects surveyed each year and in which months can be found in the (Tables S3 and S4, respectively). Line-point intercept transects were used to

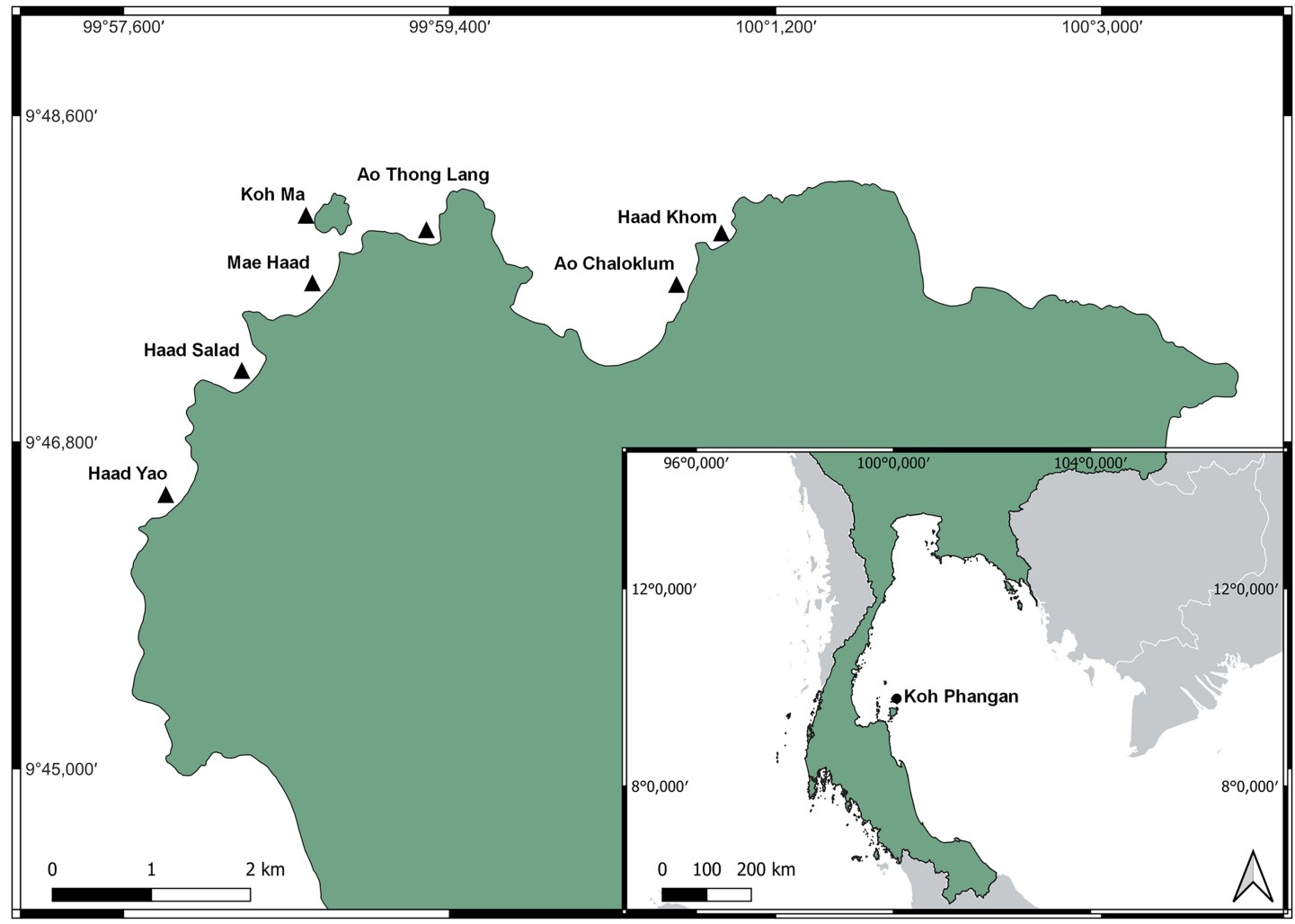

**Figure 1 Location of Koh Phangan in the Gulf of Thailand (small map) and locations of the seven surveyed reefs around the North-west coast of Koh Phangan (large map).** Basemap was provided by QGIS, which is under the Creative Commons Attribution-ShareAlike 3.0 license (CC BY-SA: https://creativecommons.org/licenses/by-sa/3.0/). The shapefile of Thailand was taken from https://data.humdata.org/dataset/cod-ab-tha, available under the Creative Commons Attribution 3.0 IGO license (CC BY 3.0 IGO: https://creativecommons.org/licenses/by/3.0/igo/).

visually study the benthic communities as they were the most cost-efficient method while showing the same precision and accuracy as the comparable belt and line transects (*Nadon & Stirling, 2006*). Consequently, for each survey, three 50-m transect lines with a 2 m gap between each other were laid out on the reef crest. SCUBA divers took a photo directly above the transect line at each 50 cm mark. This made for a total of 101 photos per transect, which summed up to 303 photos per survey and covered a distance of 150 m. The photos were analyzed by trained students in the COREsea laboratory. In each photo, the organism or substrate exactly below the respective 50 cm mark of the transect line was identified and recorded. Corals were identified to the genus level using the Coral Finder identification guide (*Kelley, 2016*) and the Corals of the World website (*Veron et al., 2016*), with the dataset being recently revised given the advances and changes in coral taxonomy
over the last years. Identification was carried out by two observers that collected the data that day, and everyone involved received coral identification training and supervision by COREsea staff. Identified hard corals were classified as live coral cover (LCC). Furthermore, turf algae overgrowing old or newly dead coral (TA) and fleshy macroalgae were recorded. Other live forms, such as anemones, soft corals, sponges, or polychaetes, were classified as other invertebrates. Sand, rocks, and coral rubble were classified as sand. After identification, the counts of each substrate type were used to estimate their respective coverage. Due to their very low presence (<0.5%), macroalgae were left out of the analysis. To distinguish between macroalgae and turf algae, fleshy macroalgae were identified as having distinguishable structural features such as fronds, stalks, and holdfasts. The four main genera of macroalgae present in the study area were Turbinaria, Padina, Lobophora, and Sargassum. Turf algae on the other hand were identified as having no distinguishable structural features and mainly represented assemblages of filamentous and/or juvenile algae that encrusted the substrate less than 2 cm in height. For better comparability with other studies and to account for possible misidentifications, the coral genera were categorized into four groups according to their growth form, which were as follows: massive, branching, plate-like, and solitary (Table S1). We also provide a description of bleaching patterns in the area, both for all corals and for each growth form group (Figs. S5 and S6).

## Statistical analysis

All data analyses and graphical representations were performed in R (Version 4.1.2, Nov. 2021). We excluded data from 2011-2013 and 2020 from the statistical analysis due to low sampling size ($n < 15$) in these years. Nevertheless, figures including all sampling years can be found in the (Figs. S3 and S4). The average benthos coverage data was calculated for each reef in each surveyed year and then combined to a yearly average for the whole island. This was done to account for missing data points because not all reefs were surveyed at the same frequency and some reefs were not surveyed in some years at all. To account for pseudoreplication due to repeated sampling of the same reefs over time, we used mixed models, specifying "reef" as a random term in each model. To test the effect of time on the LCC and TA we used a linear mixed model (LMM), while we used a generalized linear mixed model (GLMM) with a gamma error distribution and logarithmic function to test the effect of time on the invertebrates, sand, and growth forms coverage. This was done using the packages "lme4" (*Bates et al., 2015*) and "afex" (*Singmann et al., 2012*). As an indicator of biodiversity, genus richness was calculated for each year by summing up all observed genera in a given year for all reefs and calculating the mean value per year. LMM was used to test the effect of time on the genus richness of the island to identify whether there were significant changes over the years. We reported the p-values obtained from an ANOVA type III and type II for LMM and GLMM analyses, respectively, considering results to be significant with a p-value lower than 0.05 ($p < 0.05$). All raw data is available on GitHub (https://github.com/core-sea/LTMP_data).

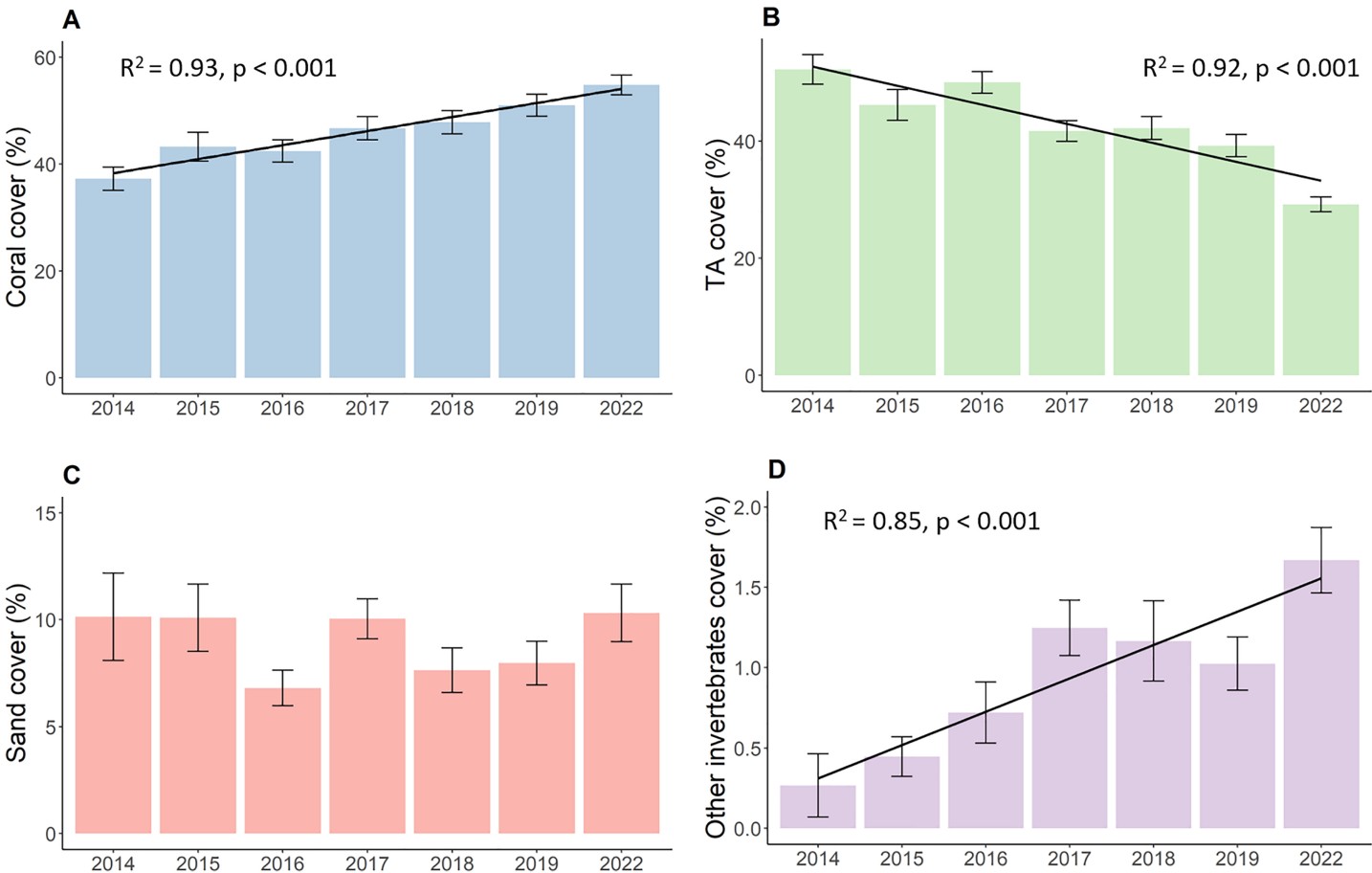

**Figure 2 Development of the benthos cover from 2014 to 2022.** (A) Live coral cover, (B) turf algae, (C) sand cover, and (D) other invertebrates cover. Bars represent the mean and error bars the standard error of the cover in the study area, while black lines represent the linear regression lines.

## RESULTS

### Development of benthic reef communities

Results show increased live hard coral cover and decreased turf algae in reefs around the northwest coast of Koh Phangan. The hard coral cover (Fig. 2A) significantly increased during the observation period from its lowest value of 37.3 in 2014 to 54.8% in 2022 ($p < 0.001$, F = 16.37, df = 679.69). This corresponds to an overall annual increase in live hard coral cover of 2.2%. The cover of turf algae (Fig. 2B) significantly decreased from 52.2% in 2014 to 29.2% in 2022 ($p < 0.001$, F = 40.04, df = 680.27), constituting a mean annual decrease of 2.9%. Yet, this decrease mainly comprised of two bigger declines in TA. The first happened from 2016 to 2017, when TA dropped from 50.0% to 41.7%. This proportion declined further up until 2019, but only in marginal steps. The second big decrease happened during the 3-year period from 2019 to 2022 where TA dropped from 39.2% to its final proportion of 29.2%. The coverage of sand (Fig. 2C) remained stable at around 9% over time with minor fluctuations while the coverage of other invertebrates (Fig. 2D) increased from 0.3% to 1.7% ($p < 0.001$, Chisq = 25.52, df = 6).

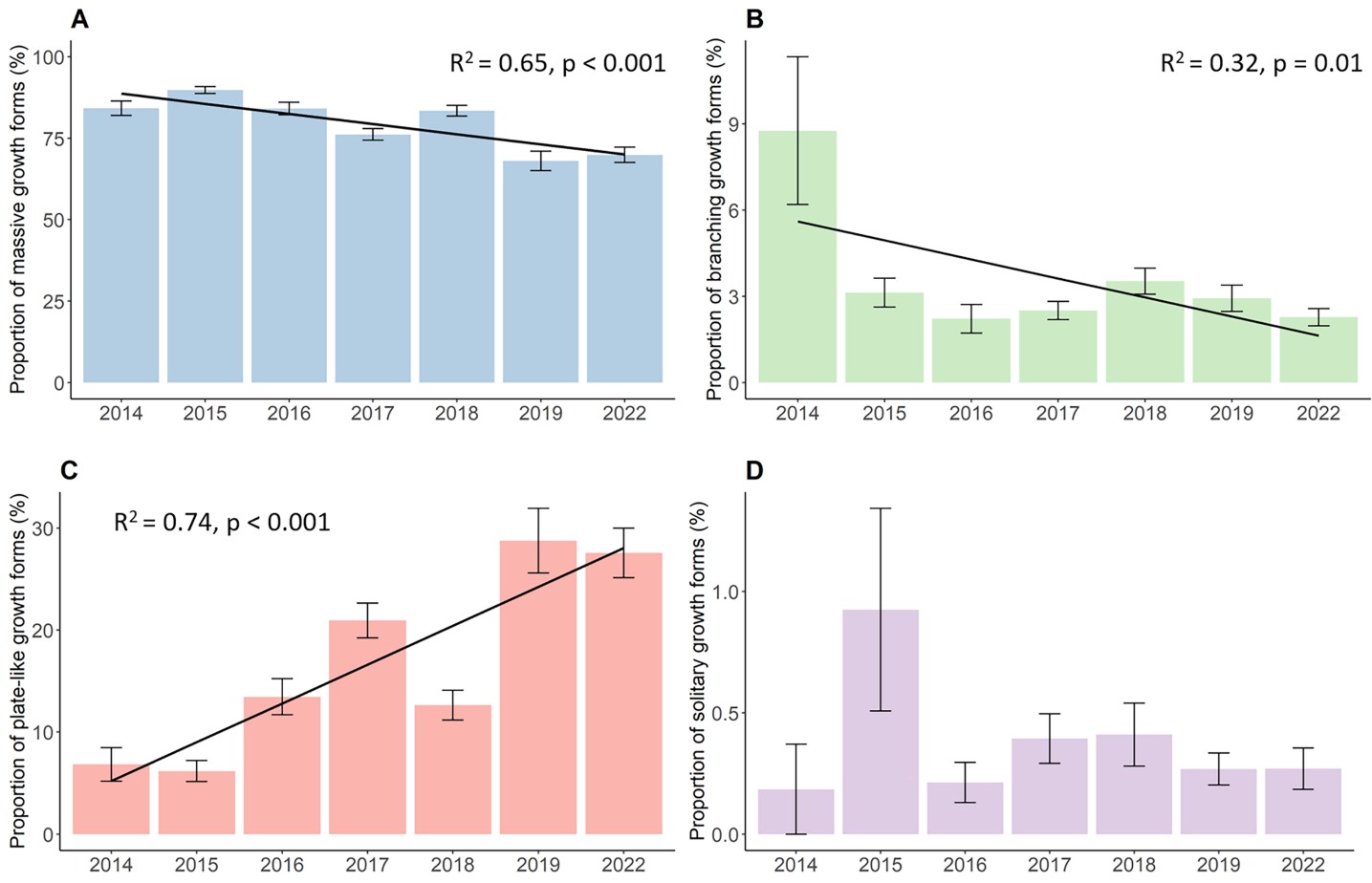

**Figure 3 Percent coverage of growth forms from 2014 to 2022.** (A) Massive growth forms, (B) branching growth forms, (C) plate-like growth forms, and (D) solitary growth forms. Bars represent the mean and error bars the standard error of the cover in the study area, while black lines represent the linear regression lines.

## Development of hard coral growth forms

The proportion of the four growth forms comprising of the total live coral cover each year also changed significantly during the whole study period (Fig. 3). Massive corals (Fig. 3A), such as *Porites*, *Galaxea*, *Alveopora*, *Dipsastraea* or *Favites*, were the most dominant growth forms throughout the observations. Although those corals were most common on reefs, they declined significantly by a total of 14.3% decreasing from 84.2% in 2014 to 69.9% in 2022 ($p < 0.001$, Chisq = 133.63, df = 6). Yet, the proportion of massive corals first increased from their initial 84.2% in 2014 to 89.8% in 2015, after which the proportion started to continuously decline down to 69.9%. Branching growth forms, such as *Acropora*, *Pocillopora*, or *Stylophora*, and plate-like growth forms, such as *Montipora*, *Turbinaria*, *Pachyseris*, or *Pavona*, showed a contrasting development with branching corals (Fig. 3B) significantly declining from 8.8% to 2.3% ($p = 0.01$, Chisq = 16.81, df = 6) while plate-like corals (Fig. 3C) quadrupled from 6.8% to 27.6% over time ($p < 0.001$, Chisq = 83.28, df = 6). Branching coral cover declined drastically within the first year down to 3.1% and further decreased down to 2.3% in the last year of the survey. Plate-like corals on the other side first decreased slightly by 6.8% in 2014 to 6.2% in 2015, and then increased up to

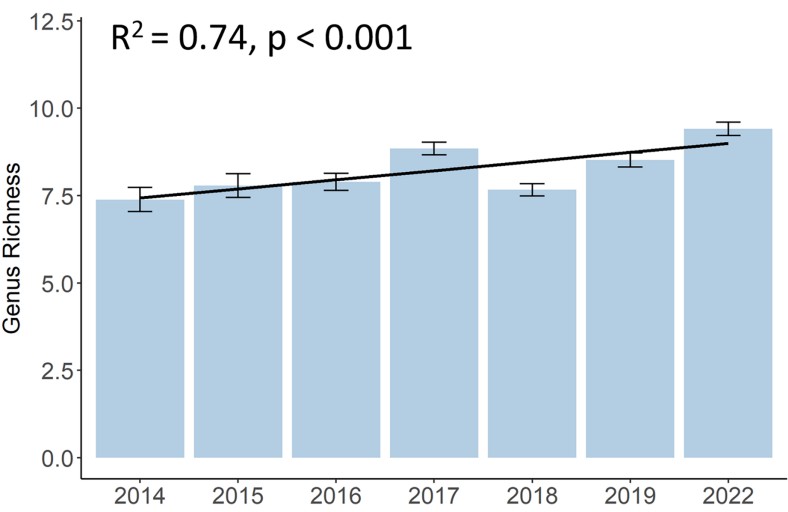

**Figure 4 Development of hard coral genus richness of Koh Phangan's coral reefs from 2014 to 2022.** Bars represent the mean and error bars the standard error in the study area, while the black line represents the linear regression line.

20.9% in 2017. The year 2018 then showed a steep drop in plate-like coral cover down to 12.6%. Plate-like coral cover then rose again in 2019 to 28.7%, where their cover stayed rather stable until 2022. The coverage of solitary corals (Fig. 3D), such as *Fungia* and *Ctenactis*, remained between 0.2% in 2014 and 0.3% in 2022, showing high standard errors and fluctuations from year to year.

### Development of hard coral genus richness

Although there was a decrease in 2018, the genus richness significantly increased from 7.4 different hard coral genera in 2014 to 9.4 in 2022 ($p < 0.001$, F = 10.12, df = 683.01) (Fig. 4). A list of all coral genera and their presence per year can be found in the (Table S2). Some genera, such as *Acropora*, *Dipsastraea*, *Favites*, *Goniopora*, *Montipora*, *Porites*, and *Turbinaria*, were present each year. Yet, genera like *Galaxea*, or *Merulina* became present only from 2015 onwards, *Cyphastrea* from 2016, and *Hydnophora* or *Pectinia* from 2017 onwards. Other genera, like *Euphyllia* were not present anymore in 2022 and other genera, such as *Stylophora* or *Ctenactis*, were sporadically observed in different years.

## DISCUSSION

Numerous studies, both recent and in the past, have demonstrated that coral reefs worldwide are facing threats such as pollution, overfishing, and rising water temperature. These factors pose a significant danger to these ecosystems and the species they support (*Bruno & Selig, 2007*; *Hoegh-Guldberg, 2011*; *Zaneveld et al., 2016*). One of the first indications of an unhealthy or degraded coral reef is the loss of living corals or the reduction of live hard coral coverage. In this study, we show that the coral cover of Koh Phangan's reefs increased between 2014 and 2022 and is mainly driven by plate-like corals. The described increase is rather surprising, considering the reported global and local declines of coral cover.

**Table 1 Selection of reported reef recoveries from different coral reef regions.**

| Location | Recovery (% yr$^{-1}$) | Publication | Recovery period |
|---|---|---|---|
| GBR | 0.8–4.0 | Lourey, Ryan & Miller (2000) | 1985–1997 |
| GBR | 0.9–5.8 | Cheal et al. (2017) | 2011–2015 |
| GBR (north) | 4.6 | AIMS (2022) | 2017–2022 |
| GBR (central) | 6.3 | AIMS (2022) | 2019–2022 |
| NW Australia | 2.9 | Gilmour et al. (2013) | 1998–2010 |
| Seychelles | 1.8 | Robinson, Wilson & Graham (2019) | 2005–2014 |
| Bonaire | 2.0 | Steneck et al. (2019) | 2011–2017 |
| Moorea | 1.0–12.0 | Holbrook et al. (2018) | 2011–2015 |
| Galapagos Island | 2.6 | Glynn et al. (2015) | 2007–2012 |
| Mexican Caribbean | 0.45 | Contreras-Silva et al. (2020) | 2005–2016 |
| Thailand | 2.2 | This study | 2012–2022 |

**Note:**
GBR, Great barrier reef.

## How did the benthic reef communities, particularly hard coral cover, around Koh Phangan develop since 2014?

During the observed timespan, the live hard coral cover significantly increased by 17.5% up to a final proportion of 54.8%. In contrast, the benthic cover of turf algae significantly decreased by a total of 23.0% down to 29.2%. When separating the benthic reef communities and their development over time by reef site, the overall trend was consistent between all sites except Mae Haad 4, where hard coral cover decreased while TA cover remained stable (Fig. S1). This difference could be explained by the fact that Mae Haad 4 is heavily affected by pulse sedimentation events, while the other sites are mainly influenced by mechanical damage.

Although the studied reefs around Koh Phangan overall showed an increase in live coral cover, the latest Status of Coral Reefs of the World report (The Global Coral Reef Monitoring Network, 2021) stated a decline of global average hard coral cover from 33.3% in 2009 to 28.8% in 2018. The mean live hard coral cover also declined in the East Asian Seas region, which includes the Gulf of Thailand, from 40.8% in 2009 to 36.8% in 2019. The annual increase of live hard coral cover of 2.2% found in this study is in line with the recovery rates reported for many other reef ecosystems around the world (Table 1). But only reefs in marine protected areas or those in remote locations showed higher recovery rates than in the present study. Reefs that are subject to higher anthropogenic influences, such as reefs in the Great Barrier Reef and the Caribbean, showed lower recovery rates. Interestingly, while the increase in hard coral cover in the present study was mainly driven by plate-like coral species, in other regions like the GBR (AIMS, 2022), Seychelles (Robinson, Wilson & Graham, 2019), and Moorea (Holbrook et al., 2018) the observed increase was attributed to the increase in branching corals like the genus Acropora and Pocillopora.

Phongsuwan et al. (2013) reported significant losses of coral cover in the Andaman Sea and the Gulf of Thailand due to the 2010 bleaching event, compared to coral cover over

**Table 2 Selection of long-term data sets from Thailand.** Shown are either increases (lower to higher cover %) or decreases (higher to lower cover %) in coral cover.

| Location | Change in coral cover (%) | Publication | Monitoring period |
|---|---|---|---|
| Kut Island, Northern GoT | 50.6 to 39.1 | *Yeemin et al. (2013a)* | 2007–2012 |
| Koh Tao, Eastern GoT | 36.2 to 39.1 | *Scott et al. (2017)* | 2006–2014 |
| Sichang Islands, Pattaya & Sattahip region, Eastern GoT | 55.6 to 62.3 | *Poosuwan (1999)* | 1995–1998 |
| Samui Island, Western GoT | 21.6 to 27.3 | *Yeemin et al. (2013b)* | 2004–2010 |
| Samui Island, Western GoT | 21.0 (only 2011 data) | *Sutthacheep et al. (2013)* | 1998–2010 |
| Phuket, Andaman Sea | 8.9 to 57.5 | *Dunne et al. (2021)* | 1997–2019 |
| Racha Yai Island, Andaman Sea | 17.5 to 24.9 | *Jaroensutasinee, Somchuea & Jaroensutasinee (2020)* | 2013–2019 |

**Note:**
GoT, Gulf of Thailand.

previous years dating back to 1988. Coral cover of most of the reefs in the middle and northern Gulf of Thailand decreased below 25%. In a more recent study, *Sutthacheep et al. (2019)* reported a mean coral cover of above 40% for most reefs in the southern Gulf of Thailand, which is similar to a coral cover of 51% in 2019 in our study. Other studies working with long-term data sets from Thai waters reported both increases and declines in coral cover (Table 2). The studies from neighboring islands such as Koh Samui (*Sutthacheep et al., 2013*; *Yeemin et al., 2013b*) and Koh Tao (*Scott et al., 2017*) showed comparable increases in coral cover to our study. The similarities of our measured values and the reported values from other parts of the Gulf of Thailand may show that the positive trend of recovering reefs in Koh Phangan is not only a local phenomenon but could be an indicator for the situation in the whole Gulf of Thailand. Also, the reefs in north-western Koh Phangan showed high recovery rates without being in a protected area, giving hope that even reefs exposed to tourism and boat traffic are able to recover if other stressors are kept low.

### How did the hard coral communities, in terms of growth forms and diversity, change over time?

Branching and plate-like growth forms showed opposite developments with branching corals significantly declining from 8.8% to 2.3% and plate-like corals quadrupling their presence from 6.8% to 27.6% until 2022. The proportion of massive corals, though slightly decreasing over time, remained the dominant growth form throughout the observed sampling period. Additionally, genus richness significantly increased by two genera until 2022 with a mean genus richness of 9.9 per transect.

The genus richness was chosen as an indicator of the hard coral diversity of Koh Phangan's reefs. We decided to use genus over species richness as data were only available on the genus level. As it is not always possible to identify the species of a coral without molecular methods, this approach posed as a robust option to determine changes in the hard coral diversity from the data available. Moreover, the influence of misidentification is lower than in those diversity indices that incorporate the abundance of genera or species. The observed trend, points towards an increased diversity of hard corals around Koh

Phangan. This not only shows that corals started to grow, potentially recovering from the 2010 bleaching event, but that they occupied new spaces and that the different reef patches became more diverse. Massive corals such as *Porites* are often slow growing with low recruitment rates (*Page, Muller & Vaughan, 2018*) and therefore new open spaces might have been colonized more often by plate-like and branching corals leading to an increase in the community proportion of plate-like corals in our study area. This observation is rather unusual as the diversity of other reefs is declining (*Bellwood et al., 2004*). Nevertheless, as more open space became available after the decrease of turf algae, this space may have been occupied by rarer species, helping to further increase habitat complexity, and therefore leading to a positive feedback loop of a recovering reef.

The dominance of massive corals has been observed in other reefs in the Gulf of Thailand as well. *Aunkhongthong et al. (2021)* and *Sutthacheep et al. (2022)* described *Porites* to be the most dominant genus in their study areas in the Western Gulf of Thailand. In our study, we found that plate-like corals, especially of the genus *Montipora*, quadrupled their proportions on the reef while branching growth forms are declining to a point of making up only 2.3% of the live hard coral cover proportion. In different reefs, like the GBR (*AIMS, 2022*), Seychelles (*Robinson, Wilson & Graham, 2019*), and Moorea (*Holbrook et al., 2018*), the observed recovery of coral reefs was mainly due to "weedy", fast-growing branching coral species from the genus *Acropora* or *Pocillopora*. Also, shifts from reefs dominated by *Acropora* towards dominance by *Pocillopora*, *Stylophora* or *Porites* have already been observed on some reefs around the world, often related to mass bleaching events and Crown-of-Thorns starfish predation (*Carlot et al., 2020*; *Edmunds, 2018*; *Riegl, Berumen & Bruckner, 2013*). A study by *Brown et al. (2019)* described a similar shift from a reef dominated by *Acropora* in 2007 to the dominance of *Pocillopora* in 2016 in Phuket. While not analyzed yet for Koh Phangan in particular, this shift in the coral community could lead to significant changes in the associated reef fauna, especially fish species abundance and composition. This could then further affect ecosystem functioning and ecosystem services that people along the coast benefit from, such as source of income (*Hoegh-Guldberg, 1999*; *Moberg & Folke, 1999*) and livelihoods (*Wilkinson, 2004*).

### Bleaching patterns in different coral growth forms

Besides the effect coral morphology has on ecosystem properties and associated fauna, coral morphology can also significantly affect corals bleaching susceptibility. It has been hypothesized that branching corals are more susceptible to bleaching and bleaching-related mortality than massive corals (*Gleason, 1993*; *Hoegh-Guldberg & Salvat, 1995*; *McCowan, Pratchett & Baird, 2012*). Yet, studies have reported differing results regarding the effect of coral morphology on bleaching patterns, such as no clear effect (*Williams et al., 2010*), higher susceptibility of branching and plate-like corals compared with (sub-) massive ones (*Loya et al., 2001*; *Marshall & Baird, 2000*; *McClanahan, 2004*), and higher bleaching susceptibility of massive compared to branching corals (*Ortiz, Gomez-Cabrera & Hoegh-Guldberg, 2009*). In summary, the patterns of bleaching susceptibility based on morphology are not as clear as previously suspected.

An in-depth analysis of bleaching patterns could shed light on whether massive and branching corals were more affected by rising temperatures on the study sites than plate-like corals. We can say that in our surveys massive corals showed the highest amounts of bleached colonies in all years. This might hint at them being more susceptible to higher temperatures than the other growth forms. Yet, it might also be biased as massive corals make up most of the live coral cover, hence it is more likely to find bleached massive colonies along the transect, than bleached solitary corals which are barely found (see Figs. S5 and S6). Without having specifically collected this data as a parameter in the surveys, personal communication with the Research manager of COREsea provided anecdotal information that during bleaching events most of the colonies that were fully bleached tended to be smaller colonies. While *Done (1999)* hypothesized that increased bleaching frequencies may shift the community towards a dominance of smaller, less-fecund coral populations, *Edmunds (2005)* predicted that coral communities would show large colony dominance on the Great Barrier Reef. So, the prediction of *Edmunds (2005)* could also hold true for the coral reefs around Koh Phangan. It may therefore be interesting to include the parameter of colony size in future surveys, to allow for a quantitative analysis of this anecdotal observation. However, this anecdotal information gives hope that the reefs around Koh Phangan did not shift toward a smaller, less-fecund coral population but instead, sustained fecund coral colonies. This, together with the high genus richness and abundance of plate-like corals are hopeful signs of a reef being able to keep and even build new structural complexity on the reef with microhabitats, possibly increasing the survival chances of corals during thermal stress.

## Did the reefs around Koh Phangan undergo shifts from coral to algae dominance or *vice versa*?

The increase in coral cover, along with the decrease in turf algae since 2014, suggested a recovery back to hard coral dominance, a so-called "reverse" phase shift. In 2014 the reefs were dominated by turf algae, possibly still as an after-effect of the 2010 bleaching event, but in 2019 live hard coral cover reached over 50% for the first time, which was further extended until 2022 with a final coral coverage of 55%. A similar reversed phase shift was described by *Idjadi et al. (2006)* for a Jamaican reef. The degraded reef transformed back into a coral-dominated state within a time span of 9 years. The authors attributed the rapid shift to the occurrence of massive *Montastrea annularis* corals. Their longevity and structure increased the complexity of the otherwise barren habitat and therefore helped the reef to recover (*Idjadi et al., 2006*). The high genus richness around Koh Phangan could have provided the same structural base that supported the further recovery of the local reefs. Yet, structural complexity is not the only component needed for a reverse phase shift. Macroalgal browsing fishes can aid in and facilitate reverse phase shifts, depending on the spatial and temporal extent of the disturbance (*Puk, Ferse & Wild, 2016*). Besides this, the amount of juvenile corals and herbivorous fishes, as well as the water depth of the reef are crucial for the trajectory of a coral reef after a bleaching event (*Graham et al., 2015*). Hence, the genus richness, as well as the habitat complexity due to different coral growth forms

around Koh Phangan most likely aided the rapid reef recovery during the observed time period by supporting coral recruitment (*Hata et al., 2017*; *Roth et al., 2018*).

## Outlook

This study found that the coral reefs of Koh Phangan have recovered in the past decade, despite challenges such as climate change and pollution. This is important because coral reefs support around half a billion people (*Camp et al., 2018*; *Hoegh-Guldberg, 2011*; *Wilkinson, 2004*), and understanding and protecting them is crucial. The data shown in this study, which was collected consistently over time, is especially valuable because it can be used to draw more accurate conclusions about long-term changes and dynamics with less bias. To better understand and predict changes in the coral reef community, it is important to continue monitoring the benthic community, environmental parameters, and other biotic factors such as fishes and coral predators. This knowledge should inform policymakers and be used to inform management strategies for coastal areas in the case of bleaching events or other factors that could impact the health of the reef.

## ACKNOWLEDGEMENTS

Thank you to all the COREsea students over the years between 2012 and 2022 who collected all the data in the field and built this unique long-term monitoring dataset. We would also like to express our gratitude to our co-author and friend, Eike Schönig, who passed away in 2019. Together with his wife Janina, he founded and operated COREsea and initiated the monitoring program. He has been an inspiration and a role model for young researchers from around the world. We miss him. COREsea's long-term monitoring program and surveys are conducted under a Memorandum of Understanding with the Department of Marine & Coastal Resources Thailand.

## ADDITIONAL INFORMATION AND DECLARATIONS.

### Funding

The authors received no funding for this work.

### Competing Interests

The authors declare that they have no competing interests.

### Author Contributions

- Florian Stahl performed the experiments, analyzed the data, prepared figures and/or tables, authored or reviewed drafts of the article, and approved the final draft.
- Selma D Mezger analyzed the data, prepared figures and/or tables, authored or reviewed drafts of the article, and approved the final draft.
- Valentina Migani analyzed the data, authored or reviewed drafts of the article, and approved the final draft.
- Marko Rohlfs analyzed the data, authored or reviewed drafts of the article, and approved the final draft.

- Victoria J Fahey conceived and designed the experiments, performed the experiments, prepared figures and/or tables, authored or reviewed drafts of the article, and approved the final draft.
- Eike Schoenig conceived and designed the experiments, performed the experiments, authored or reviewed drafts of the article, and approved the final draft.
- Christian Wild analyzed the data, authored or reviewed drafts of the article, and approved the final draft.

## Data Availability

The raw data is available at GitHub and Zenodo:

- https://github.com/core-sea/LTMP_data.

- Victoria Fahey. (2023). core-sea/LTMP_data: COREsea longterm monitoring data (v1.0.0). Zenodo. https://doi.org/10.5281/zenodo.7964564.

## Supplemental Information

Supplemental information for this article can be found online at http://dx.doi.org/10.7717/peerj.16115#supplemental-information.

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
