# Peer review of "Recent and rapid reef recovery around Koh Phangan Island, Gulf of Thailand, driven by plate-like hard corals"

_PeerJ, doi:10.7717/peerj.16115_

## Round 0.1 · original submission · Major Revisions

The manuscript provides some valid information with long-term data on coral reefs. The highlight is authors documented long-term coral reef trends in Thai waters, however, the literature needs to be updated.

We received comments from 3 reviewers and I advise the authors to follow all 3 reviewers’ comments carefully and address the issues. The authors overlooked to refer many references in relation to this work, so please follow the reviewers' suggestions of including those references and rewrite the method section effectively for the readers to understand. English needs to be checked according to a reviewer. Based on all reviewers' comments, I suggest major revision.

Reviewer 1 ·

Basic reporting

Introduction and background are clear and well-referenced.
Research question is well-defined, relevant, and meaningful.

Indeed, this study is not the first to compare the long-term change in coral cover in the Gulf of Thailand using data collected by a single organization, using a consistent survey method and fixed sites at regular intervals over several years (as stated on lines 80-82). The fact is that there are several long-term monitoring data in Thailand. The authors can read more in the following publications for use in the literature review and comparison in the discussion section.
• Report of observation and assessment of marine and coastal resources; coral reefs and seagrass beds, fiscal year B.E. 2563 (downloadable at https://www.dmcr.go.th/detailLib/6728). Unfortunately, this report is in Thai. However, the graphical histogram of live coral cover over time is quite easy to understand. The graphs, fig. 78 and 79, are depicted from several sites at Pha-ngan Island from 2005 to 2020. Note that the present data (2022) is also available, but not yet published. The authors might also use the data from other nearby areas (such as the Samui Islands group from this report) or other places in Thailand, either in the Gulf or the Andaman Sea, for comparison in the discussion.
• Long-term changes in coral communities under stress from sediment. Deep-Sea Research II 96 (2013) 25–31 (Downloadable at https://www.sciencedirect.com/science/article/abs/pii/S0967064513001586?via%3Dihub). This work describes the change in live coral cover and coral community over time (2004 - 2010) at Samui Island in particular.
• Impacts of the 1998 and 2010 mass coral bleaching events on the Western Gulf of Thailand. Deep-Sea Research II 96 (2013) 25–31 (downloadable at https://www.sciencedirect.com/science/article/abs/pii/S0967064513001574?via%3Dihub).
• Long-term decline in Acropora species at Kut Island, Thailand, in relation to coral bleaching events. Mar Biodiv (2013) 43:23–29. This work describes the change of Acropora in particular, during 2007 – 2012, at Kut Island.
• Repeated coral bleaching in the Andaman Sea, Thailand, during the last two decades. Phuket mar. biol. Cent. Res. Bull. 71: 19–41 (2012), (downloadable at https://www.dmcr.go.th/dmcr/fckupload/upload/44/image/FullpaperPMBC/2012%20Vol.71%20Phongsuwan%2019%2041.pdf). This work shows the change of coral cover and the phase shift of coral community, during 1990 - 2010 (see the example in the histogram fig. 6 and 7), in the Andaman Sea in particular.
• The recent history of coral reefs and their environment in Tang Khem Bay, Phuket, Thailand – An example of corals living in a potential climate change refuge? Phuket mar. biol. Cent. Res. Bull.76: 25–39 (2019), (downloadable at https://www.dmcr.go.th/ckeditor/upload/files/id138/full%20paper/2019_Vol.76%20brown%2025%2039.pdf). In fig. 5 B, there is the example of the phase shift, Pocillopora becomes replacing the reef. This might be put as a reference in the text, lines 268-270.

Experimental design

In methodology, the authors used Line-point intercept transects to study the benthic communities. It is not so clear in the explanation of the method (so the reader might have to search and read the reference of Nadon and Stirling (2006)). The authors described that the “substrate exactly below the respective intercept point of the transect was noted”. This is not understandable. It is more favorable if the authors briefly explain how to pick up the data from each photo and transform the data into the percentage cover of each type of substrate.

Validity of the findings

As this study interprets the community of coral in terms of genera richness, it would be very useful if the authors also have a Table of a list of coral genera encountered on the transects each year.

The authors interpret the data of all sites in average value (such as coral cover value). This is acceptable. However, if there is an interpretation in addition, with site separation as well, this will help to understand the human impact from each site. (In the introduction part, lines 99-109, the authors described that there are several types of human impact among the study sites). The results would be beneficial for management.

Figure 2; Is it possible to plot the regression line over the histogram of fig. A and B in order to show the trend of change of cover of coral and turf algae in particular?

Additional comments

The language is well-interpreted. However, there might be some errors in words or sentences. The authors may need English proved by the English mother tongue reviewer.
All scientific names, i.e. genus name of the coral must be in italic form.

Figure 1; Map of Pha-ngan Island needs to put the latitudes and longitudes on the frame. This will help the readers locate the place more easily.

I admire the authors for gathering long-term data and writing this work. It would be useful for coral reef management in that region where tourism impact is increasing.

Reviewer 2 ·

Basic reporting

- The authors should provide scientific names of the corals in their text for better understanding of the readers.
- Some references of coral reef studies in the western Gulf of Thailand should be added.

Experimental design

- The survey method in this study is different from the standard methods used in this region, such as English et al., 1997.

Validity of the findings

- The findings are generally good for managing coral reefs in the Gulf of Thailand. However, it is better if the authors provide scientific names of the corals.

Additional comments

- The authors should provide more details of the results.

Reviewer 3 ·

Basic reporting

The language used in this manuscript is on the whole clear and concise. The authors have made some effort to support many of their points with valid and contemporary literature, however with much room for improvement. In a few cases (highlighted in detail below) the supporting literature is inappropriate either due to the weak link between the point being made and reference, or entirely erroneous where the reference does not support the point being made. A far greater weakness is the relative paucity of contemporary or historic literature from the region of interest, with relatively little mention or discussion using studies from Thailand or the Gulf of Thailand. Much of the literature included is used to compare other regions with Koh Phangan, with little mention of contrasting findings within the Gulf of Thailand. Overall, the figures provided are well described with much off the relevant raw data supplied as supplementary. However crucially, the survey-specific or number of surveys per year data is not available in the supplementary data, which arguably corresponds to the true raw data. A simple table providing findings per survey per date would constitute a correct reporting of raw data and allow reviewers to investigate inferences further.

Experimental design

The present study aims to provide a contemporary insight on shifting trends of coral reefs from Koh Phangan. The research questions are outlined broadly at the end of the introduction and are within the scope of the journal. As such, the methodology of line-intercept surveys across the multi-year period is an appropriate methodology to provide a basic overview from the specific sites surveyed. It should be noted that the findings have been generalised (both spatially and temporally) far beyond the suitability of the data collected. The methodology has been provided to sufficient detail such that replication is theoretically possible, though may be improved significantly by more details on taxonomy (i.e improved identification of algae and corals) and number of replications.

Validity of the findings

The present study has applied an appropriate methodology to assess the important research questions outlined. Many of the findings provided such as diversity and algal cover provide valuable insights on the status and trends at specific reefs at Koh Phangan, which may indeed have utility in a broader understanding of trends at Koh Phangan. However, many of the most significant trends can be attributed to data differences between 2012 and 2013, suggesting either a significant ecological shift between the years 2012 and 2013 (not reported in other studies), or by a weakness in methodology. The theme of significant coral reef recovery is somewhat dubious based on the data provided, though it is evident that some amount of growth, if not stability, is present in the studied reefs. The claim that plate-like corals are responsible for much of the community structure change seen appears to be well-supported.

Additional comments

Overall, this study represents an important contribution to the documentation of coral reef trends in Thai waters and the Gulf of Thailand specifically, based on data from an island that has been under-studied for a number of years. However, I have two general criticisms on the presented manuscript, which I have elaborated on in the detailed comments below. The first is an unfortunate lack of comparison between the findings presented here and those elsewhere from Thai waters. This study erroneously claims to be a first in many regards, and the comments throughout compare the presented findings with distant reefs without eluding to most of the other research historically conducted in the Gulf of Thailand. For example, while the authors have acknowledged the Department of Marine and Coastal Resources, they have neglected to mention their publicly available reports that provide extensive analyses from coral surveys across the Gulf of Thailand. Secondly and arguably more significantly, the authors document significant trends over the decade surveyed, many of which appear to rely on the contrasting dataset collected in 2012. The authors should attempt to explore the cause of such a drastic (and in many cases, unlikely) shift, and provide more details of the survey intensity in 2012. Alternatively, if the dataset from 2012 is indeed found to be weaker (as suggested in lines 138-140), the authors should consider re-analysing the trends omitting the 2012 dataset. Many of the results, and Figures 2 and 4 do highlight just how distinct the 2012 dataset is compared with the rest.

Detailed comments:
• Lines 64 – 67: The authors suggest that Phongsuwan et al. demonstrated that offshore reefs were less impacted and threatened than onshore reefs. Firstly, this is manifestly not what this paper shows, with significant damage reported from offshore sites. Secondly, this paper looked at 5 offshore regions and only a single near-shore, hardly allowing for comprehensive estimation to support such a claim.
• Lines 80 – 82: The authors claim the submitted manuscript as being the first to show long-term changes in coral cover using data by a single organisation from the Gulf of Thailand. The first and most important fact is that long-term monitoring over permanent transect sites go as far back as 1980 if not earlier across both the Andaman and Gulf of Thailand (see references below). Indeed, long-term monitoring data from the nearby island of Koh Tao has been available for a number of years suggesting the present manuscript presents not even the first such case from the Samui archipelago. While I certainly agree that long-term monitoring efforts are essential for a deeper understanding of shifting reef communities, and the present manuscript offers valuable information for Koh Phangan, the authors must be careful when inflating the importance of certain findings or data. The second point I must raise is to question why data collected by ‘a single organisation’ is considered to of particular value? The authors have not made any reference to why institutional isolation plays a part in the value of a given dataset. Below are some references of note regarding the above points.
o Chou, L.M., Sudara, S., Manthachitra, V., Moredee, R., Snidvongs, A. and Yeemin, T., 1991. Temporal variation in a coral reef community at Pattaya Bay, Gulf of Thailand. In Fourth Symposium on our Environment: Proceedings of the Fourth Symposium on Our Environment, held in Singapore, May 21–23, 1990 (pp. 295-307). Springer Netherlands.
o Chon Poosuwan, B.S., 1999. Temporal Variation in the Coral Reefs of the East Coast of the Inner Gulf of Thailand (Doctoral dissertation, Brock University).
o Phongsuwan, N. and Chansang, H., 2012. Repeated coral bleaching in the Andaman Sea, Thailand, during the last two decades. Phuket Marine Biological Center Research Bulletin, 71, pp.19-41.
o Scott, C.M., Mehrotra, R., Cabral, M. and Arunrugstichai, S., 2017. Changes in hard coral abundance and composition on Koh Tao, Thailand, 2006–2014. Coastal Ecosystems, 4, pp.26-38.
• Lines 87 – 89: The authors support the rather significant generalisation that high coral cover and high diversity usually show greater ‘resilience’ by a study focusing on herbivorous fish in reefs. Certainly herbivory, biofouling and competition, and nutrient loading play a role in reef threats and resilience, however, the present manuscript among many others document a far wider range of threats and factors that may affect resilience. While a case may certainly be made that many sites of higher cover or diversity show lower mortality under certain threats, this is far from being a settled case. For example, many reefs in the Andaman sea of Thailand, known to support a greater diversity and higher coral cover, have undergone far greater losses in the face of bleaching and other threats, than many in the Gulf of Thailand. This sentence should be justified far better or removed.
• Line 113: The methodology states that data was collected till February 2020 but do not provide a starting month for data from 2012. The starting month of 2012 data may play a significant role in the proportion of algae documented on the reef, a feature that has strongly been linked with seasonality. Both, the number of replicates per year, and the time of the year that data is collected, may play an influential role on the proportion of algae found on reefs. It is my opinion that the 2012 dataset is the basis for some of the most dramatic claims in the present study (from shifting coral cover to changes in algal cover). The authors should state how many surveys were carried out per year, and the starting date for 2012 surveys.
• Lines 123 – 124: Did the authors use any taxonomic literature to support their coral identifications? A great many significant advances in coral taxonomy have been made since the 2016 coral finder, much of which are also not seen on the Corals of the World website. For example, the genera Dipsastrea, Paragoniastrea, Fimbriaphyllia among others are lacking from such resources, but outdated genera such as Symphyllia and Favia (in the latter case, no longer considered valid in Thai waters) continue to persist. Supplementary data provided indeed shows coral genera that are highly unlikely to be found around Koh Phangan (i.e. Seriatopora and Heliofungia) given their absence from contemporary surveys from the Gulf of Thailand or updated taxonomy (i.e. Symphyllia and Favia).
• Line 127: The authors distinguish ‘turf algae’ from ‘macroalgae’. This needs to be elaborated as it is unclear how broad or narrow the definition of turf algae is applied in this case, particularly since the authors separate it from macroalgae due to it's low % cover.
• Lines 138 – 140: The authors choose to omit benthos and richness data for 2012 due to low sampling effort of two surveys. Are the remaining claims of 2012 data based on the same two surveys? As I’ve commented above, the 2012 data is the basis for many of the most dramatic claims in this study, and if it is indeed based on a far lower or spatially distinct dataset than the remaining years, it is likely to far better explain the trends visible.
• Line 161: The authors claim that coral cover increased by 19.3%. This suggests almost a doubling of LCC in a single year (26.3% to 45.6%). This adds further doubt on the value of data collected in 2012 as being sufficiently representative.
• Line 167: The authors suggest that turf algae dropped from 66.1% to 45.2%. Was this between 2012 and 2013, or during the year of 2013?
• Line 177: It appears that the % mentioned here (75.9% in 2013) refer to the proportion of all corals as opposed to earlier references to % cover (for example, max coral cover throughout your survey period is 54.8%, far lower than the decline of massive corals to 69.9%). This needs to be specified and clarified to prevent confusion among readers.
• Lines 191 – 192: Regarding the special treatment of solitary growthforms due to them being ‘less likely to be found right underneath a sampling point’. Unless you standardised the sizes of every colony surveyed, the same sampling bias may potentially occur with every coral type. The authors do not estimate or provide proportions of colony sizes in branching species vs. plate-like species vs. massive species either, thus the individual colony size bears no specific weight on solitary corals over other growthforms. Certainly the mobility may play a role in increased inter-survey variability, however this is likewise not a feature of note given the aim of comparing proportions using the PIT method.
• Line 196: This claim indicates a doubling of coral diversity over a single decade at the surveyed sites. Are the authors suggesting that such significant recruitment of previously unestablished genera occurred within a single decade? Can the authors point to which genera were absent at the start of the study and which were newly recorded towards the end? Might the survey methodology be responsible for this change? This remarkable shift again points to the weakness in utilising data from 2012 which is responsible for vast majority of this doubling (as per Fig. 4). Without the sharp increase from 2012 to 2013, all other years show variability of no more than 2.5 or 3 across the whole survey period.
• Lines 232 – 235: This sentence appears to be based on a trend of only two other studies, neither of which make similar claims on positive trend of recovering reefs, and indeed the vast majority of other studies from Thailand continue to show the opposite. A far more balanced discussion of this point may be made by including any of the dozens of studies documenting the status of or changes in coral reefs in the Gulf of Thailand.
• Lines 241 – 246: Regarding the remarkable temporal changes documented in the study. Can the authors provide values for what the trends would look like from 2013 as opposed to 2012? How would these values be different without the single drastic change between 2012 and 2013?
• Lines 269 – 273: Again the authors choose to compare findings from regions far afield from the present study site, but make no mention of occurrences nearby such as others within the Gulf of Thailand. How might the trends observed relate to other findings within the Gulf of Thailand? Additionally, coral genus names should be italicised in the present paragraph.
o Yeemin, T., Saenghaisuk, C., Sutthacheep, M., Pengsakun, S., Klinthong, W. and Saengmanee, K., 2009. Conditions of coral communities in the Gulf of Thailand: a decade after the 1998 severe bleaching event. Galaxea, Journal of Coral Reef Studies, 11(2), pp.207-217.
o Scott, C.M., Mehrotra, R., Hein, M.Y., Moerland, M.S. and Hoeksema, B.W., 2017. Population dynamics of corallivores (Drupella and Acanthaster) on coral reefs of Koh Tao, a diving destination in the Gulf of Thailand. Raffles Bulletin of Zoology, 65.
o Yeemin, T., Pengsakun, S., Yucharoen, M., Klinthong, W., Sangmanee, K. and Sutthacheep, M., 2013. Long-term decline in Acropora species at Kut Island, Thailand, in relation to coral bleaching events. Marine Biodiversity, 43, pp.23-29.
o Sutthacheep, M., Yucharoen, M., Klinthong, W., Pengsakun, S., Sangmanee, K. and Yeemin, T., 2013. Impacts of the 1998 and 2010 mass coral bleaching events on the Western Gulf of Thailand. Deep Sea Research Part II: Topical Studies in Oceanography, 96, pp.25-31.
• Lines 289 – 325: These paragraphs for the first time accurately question whether there is indeed a link between growthform and bleaching patterns. Bleaching susceptibility has a long history of instead being shown to be associated with taxonomy (aka specific species or genera). Authors should be wary of conflating susceptibility to growthform as opposed to taxa. Several genera in the Gulf of Thailand contain species displayed highly variable growthforms (Pavona, Psammocora, Porites and others not beginning with P). In the absence of data analysis comparing genera with bleaching susceptibility (which based on the current methodology, should be possible), emphasis on the growthform link in above passages should be scrutinised.
o Guest, J.R., Baird, A.H., Maynard, J.A., Muttaqin, E., Edwards, A.J., Campbell, S.J., Yewdall, K., Affendi, Y.A. and Chou, L.M., 2012. Contrasting patterns of coral bleaching susceptibility in 2010 suggest an adaptive response to thermal stress. PloS one, 7(3), p.e33353.
o Pratchett, M.S., McCowan, D., Maynard, J.A. and Heron, S.F., 2013. Changes in bleaching susceptibility among corals subject to ocean warming and recurrent bleaching in Moorea, French Polynesia. PLoS one, 8(7), p.e70443.
o Matsuda, S.B., Huffmyer, A.S., Lenz, E.A., Davidson, J.M., Hancock, J.R., Przybylowski, A., Innis, T., Gates, R.D. and Barott, K.L., 2020. Coral bleaching susceptibility is predictive of subsequent mortality within but not between coral species. Frontiers in Ecology and Evolution, 8, p.178.

---

## Round 0.2 · Minor Revisions

Thanks for your revision. It has been improved well, but please follow 2 reviewers comments and revise accordingly. Reviewer 1 has given some important comments to improve the article, please follow it.

Reviewer 1 ·

Basic reporting

Please see item 4 below.

Experimental design

Please see item 4 below. Note the authors already described the field method more clearly at the point of the first comment.

Validity of the findings

Please see item 4 below.

Additional comments

Second-time comment
The manuscript was much better improved. There are a few comments to add as follows;
• In the abstract, the sentence “The Gulf of Thailand is a highly under-investigated region where longterm developments of coral reef communities have not yet been described” is not true. It was already commented in the first round that there are several works involved with the long-term monitoring of coral reef communities in the Gulf of Thailand. This manuscript itself has shown the longterm monitoring sites in the Gulf of Thailand in Table 2.
• In the graph Fig.2, 3, 4 the linear regression should show the r-value and p-value. In Fig 3 B and D, by naked eyes, seem like the linear regression doesn’t fit, then maybe some other kind of curve might be applied. Also please check the p-value, if higher than 0.05, then the regression line is not necessary. (please consult a statistician for this matter).
• In Table 2, selection of long-term data sets from Thailand, please carefully re-check the values if there are mistakes. For example in the Andaman Sea (Phongsuwan and Chansang, 2012), live coral cover changed from 76.5% to 74.0% during the monitoring period 1990-2010. This is weird. As I checked, there are no figures like that. That paper does not conclude for the whole Andaman Sea, but for some selected sites. Indeed there are the graphs showing percentage of live coral cover over time (1990-2010) of many sites. So, an example of one site might be used to show in this manuscript.
• Line 113, there is no need to state that all data was collected at the same time of the day between 10:00 am – 11:00 am because the tide doesn’t matter with the data-filed collection.
• Line 295-308, is not necessary. The context is deep down to the fish community which is not the main thing to discuss in this article.
• Line 310-333; the text is too long. The authors should brief it to make it concise, using the findings (references) which directly relate to what is found at Koh Phangan.

Reviewer 3 ·

Basic reporting

Improved signficantly after revision.

Experimental design

Improved signficantly after revision.

Validity of the findings

Valid and valuable for a broader understanding of coral communities within the Gulf of Thailand.

Additional comments

None.

---

## Round 0.3 · Minor Revisions

Reviewers have recommended your paper for publication after a minor revision. Please follow the Reviewer #1 has given your more comments which are missed to follow in the first revision. Please follow the comments and suggestions and revise it. Authors need to focus on statistics in addition to the English usage.

**Language Note:** The review process has identified that the English language must be improved. PeerJ can provide language editing services - please contact us at copyediting@peerj.com for pricing (be sure to provide your manuscript number and title). Alternatively, you should make your own arrangements to improve the language quality and provide details in your response letter. – PeerJ Staff

---

## Round 0.4 · accepted · Accept

Thank you for your contribution, The authors have addressed all of the reviewers' comments, but authors should check English editing. I accept this manuscript for publication.

Reviewer 1 ·

Basic reporting

Clear.

Experimental design

Clear.

Validity of the findings

Clear.

Additional comments

The revision is completed and valuable for publishing.